# Time Series Segmentation Based on Stationarity Analysis to Improve New Samples Prediction

**DOI:** 10.3390/s21217333

**Published:** 2021-11-04

**Authors:** Ricardo Petri Silva, Bruno Bogaz Zarpelão, Alberto Cano, Sylvio Barbon Junior

**Affiliations:** 1Department of Electrical Engineering, State University of Londrina, Londrina 86057-970, Brazil; 2Department of Computer Science, State University of Londrina, Londrina 86057-970, Brazil; brunozarpelao@uel.br (B.B.Z.); barbon@uel.br (S.B.J.); 3Department of Computer Science, Virginia Commonwealth University, Richmond, VA 23284, USA; acano@vcu.edu

**Keywords:** time series segmentation, stationarity analysis, time series prediction improvement, size reduction in time series

## Abstract

A wide range of applications based on sequential data, named time series, have become increasingly popular in recent years, mainly those based on the Internet of Things (IoT). Several different machine learning algorithms exploit the patterns extracted from sequential data to support multiple tasks. However, this data can suffer from unreliable readings that can lead to low accuracy models due to the low-quality training sets available. Detecting the change point between high representative segments is an important ally to find and thread biased subsequences. By constructing a framework based on the Augmented Dickey-Fuller (ADF) test for data stationarity, two proposals to automatically segment subsequences in a time series were developed. The former proposal, called Change Detector segmentation, relies on change detection methods of data stream mining. The latter, called ADF-based segmentation, is constructed on a new change detector derived from the ADF test only. Experiments over real-file IoT databases and benchmarks showed the improvement provided by our proposals for prediction tasks with traditional Autoregressive integrated moving average (ARIMA) and Deep Learning (Long short-term memory and Temporal Convolutional Networks) methods. Results obtained by the Long short-term memory predictive model reduced the relative prediction error from 1 to 0.67, compared to time series without segmentation.

## 1. Introduction

The growth of data generation increases daily due to the advancement of technology [1]. With the advent of sensors that are capable of capturing precious data, there is also the need to transform this data into information. The most common data structure in the era of automatic sensor data processing is time series. A time series can be defined as a set of sequential data ordered in time [2]. Traditionally, stochastic processes are used to model time series behavior with great success [3,4]. In addition, machine learning-based approaches are also employed to perform the identification of complex behaviors of nonlinear patterns, optimization of unconventional functions, and even establishing connections with long dependencies through recurrent neural networks [5,6]. These patterns can be verified in different areas, such as climatic data [7], sales [8], medical diagnosis [9,10,11], security [1,12], and even the change in share values on the stock exchange [13].

From time series analyses, it is possible to examine these patterns and create predictions of future samples, as discussed in Mahalakshmi et al. [14]. Models based on machine learning, e.g., Long short-term memory (LSTM) and Temporal Convolutional Network (TCN), have shown promising results, [15,16], as an alternative to statistical models. Approaches that apply machine learning concepts can adapt their settings to improve predictive ability [17]. This can be done by adjusting their hyperparameters so that the time series modeling is better suited to the data patterns. However, in addition to being a time-consuming process, there are no guarantees that these changes in the prediction model settings will be able to adjust to the patterns found in the time series. This can occur due to external factors or even problems in capturing this data, such as discussed in Breed et al. [18], which describes tracking tag procedures to maximize the use of time series data and also by [19], which makes an extensive study of time series data mining techniques. These discrepancies can be evidenced by time series analysis and need to be addressed.

We developed a hypothesis for scenarios where the temporal samples show non-stationary variability; that is, it does not manifest stability in the behavior of its samples over time with a constant average. As far as we know, no study focused on time series segmentation resorted to the use of stationary analysis. Usually, segmentation processes are based on static thresholds [20], classical techniques such as sliding window assessments [3], and even complex deep learning procedures [6]. Thus, we consider our hypothesis to be the first proposal considering stationarity to perform time series segmentation, taking advantage of this important feature when describing a time series. It is based on time series preprocessing, which can lead to improvements in new samples prediction and even a size reduction for further storage. This preprocessing consists of segmenting data from the samples that present a non-stationary behavior over time. The advantage of this approach lies in the simplicity of stationary analysis to accurately identify samples to be segmented without relying on more sophisticated techniques, such as Deep Learning methods [6].

In this work, we introduce two proposals to perform time series segmentation based on a stationary study. The first is a framework that uses the information obtained by stationary analysis to properly adjust the change detector hyperparameters, which are usually applied in Stream Mining. This change detector is responsible for determining which samples should be segmented. In the second proposal, a new change detector was developed. Only information from the stationary analysis of the time series is considered to perform the segmentation. To make the stationary analysis, the Augmented Dickey-Fuller (ADF) test [21] was adopted. The objective of these proposals is to improve the prediction of new samples in the time series. The time series tend to have a better representation with the removal of non-stationary samples. Consequently, it is easier to model and predict future samples since the training set will be better adjusted. Some approaches (Carmona-Poyato et al. [22]; Keogh et al. [3]) use time series segmentation to reduce the amount of data to be processed and to represent the data in a more compact form, or even to extract precious information for decision-making [23]. Our approach fulfills these conditions as a result of the segmentation process and opens the door for several applications, whether for classification, compaction, information extraction, or even prediction of new samples.

Different predictors were evaluated to validate the effectiveness of our approach. The first one is considered as a baseline and is known as naive, which assumes that the next sample to be predicted will have the same value as the previous sample. There is also a statistic model known as ARIMA [24], and two recent approaches based on Deep Learning (DL) techniques, LSTM [17], and TCN [25], respectively. Moreover, distinct time series databases were used. They are composed of real-file IoT databases. Altogether, they compose ten different time series datasets from three distinct origins. We have temporal data from a photovoltaic plant, daily temperature measurements over a ten-year interval, and a monthly count of the number of sunspots observed over more than 200 years.

According to the results obtained, our hypothesis can bring benefits for new sample predictions. DL techniques obtained significantly smaller relative prediction errors, such as 0.63 for the LSTM predictor, compared to the original error equal to one in the PV plant database.

The main contributions of this paper are as follows:A new proposal for time series segmentation based on stationarity, named ADF-based segmentation.A framework to perform segmentation of time series based on stationarity using change detector algorithms (e.g., Page-Hinkley (PH) and ADWIN (ADW)), called Change Detector segmentation. Additionally, three techniques to tune the hyperparameters of the change detection algorithm, called Bigger in Smaller out, Bigger in, and Smaller out.An analysis on the improvement of the predictive capacity of time series using segmentation through stationary analysis.

The work is presented with the following structure. Section 2 introduces the problem definition and how we intend to solve the segmentation process based on the stationarity analysis. Related work is discussed in Section 3, highlighting our contributions and differences to existing segmentation techniques in the literature. Section 4 introduces the methodology of the proposal of this work. Section 5 presents the experimental studies that were conducted to assess the performance of our approach compared with the state of the art. Finally, Section 6 summarizes the conclusion of this work.

## 2. Preliminaries

In this section, important theories to support the development of the work will be considered. First, the purpose of employing segmentation in time series will be presented, as well as the discussion about stationarity, which is a fundamental feature of this work.

### 2.1. Segmentation Process

The segmentation process consists of identifying heterogeneous information for a certain group, database, etc. This process can highlight misbehaving samples in time series, isolate them from the rest of the data, reduce its size, among other possible applications, which facilitate its analysis. There are some approaches based on sliding windows and specific thresholds that rely on statistic data analyses to perform the time series segmentation [3]. In this work, the segmentation process is motivated by the stationarity analysis of the time series.

**Definition** **1** (Time Series)**.**
*A time series can be defined as a set of sequential data, ordered in time [2]. It can be collected at equally spaced time points and we use the notation $y_{t}$ with (t=…,−1,0,1,2,…), i.e., the set of observations is indexed by t, representing the time at which each observation was taken. If the data was not taken at equally spaced times, we denote it with i=1,2,…, and so, $\left(t_{i}-t_{i-1}\right)$ is not necessarily equal to one [26].*

**Definition** **2** (Stationarity)**.**
*According to the definition of random processes [27], a discrete-time or continuous-time random process X(t) is stationary if the joint distribution of any set of samples does not depend on the placement of the time origin. This means that the joint cumulative distribution function of $X(t_{1})$, $X(t_{2})$, …, $X(t_{k})$ is the same as that of $X(t_{1}+\tau )$, $X(t_{2}+\tau )$, …, $X(t_{k}+\tau )$ for all time shifts τ, all k, and all choices of sample times $t_{1},\ldots ,t_{k}$.*

The stationarity of a time series can be explained according to its stability during a time with a constant average. However, in some real cases, certain trends influence the time series behavior, which ends up affecting its stationarity. A series can be stationary for short or long periods, which implies a change in the inclination of the series. The Augmented Dickey-fuller test was applied in this work to evaluate the time series stationarity.

### 2.2. Augmented Dickey–Fuller Test

The Augmented Dickey-Fuller test is a statistical analysis that verifies the null hypothesis that a unit root is present in a time series sample [21]. The null hypothesis can inform whether a given time series is stationary or not. It checks if the time series can be represented by a unit root with a time-dependent structure [28]. If this hypothesis is rejected, we can assume that the time series is stationary. In other words, the statistics are not affected by time offsets. The unit root exists when the α assumes the value one, according to Equation (Equation 1):(1)$$Y_{t}=\alpha Y_{t-1}+\beta X_{e}+\epsilon$$
where *t* represents the time, $Y_{t}$ represents the value of the time series at time *t*, and $X_{e}$ is an exogenous variable, which is explanatory and belongs to the time series. ϵ is a serially uncorrelated, zero-mean stochastic process with constant variance $\sigma^{2}$. The ADF test assumes the α value equal to one in the following equation:(2)$$Y_{t}=c+\beta t+\alpha Y_{t-1}+\phi \Delta Y_{t-1}+\epsilon_{t}$$
where *c* is a constant, β is the coefficient on a time trend, $Y_{t-1}$ represents the lag order equals one, $\Delta Y_{t-1}$ is the first difference of the series at time t−1 and ϵ is the exogenous variable.

However, the confirmation or rejection of the hypothesis can be simply verified by analyzing the *p*-value of the ADF test. A *p*-value below a threshold determines rejection of the null hypothesis, otherwise suggesting the failure to reject the null hypothesis. This representation can be verified according to the conditions given below.
(3)$$\delta =\left\{\begin{array}{cc} 1, & \mathrm{if}\;p-\mathrm{value}<\gamma \\ 0, & \mathrm{otherwise} \end{array}\right.$$
where δ represents a time series stationarity and γ represents a determined threshold. We assume the stationarity of the series if the value generated by the conditional is one. A non-stationary series is obtained otherwise.

In addition to the *p*-value analysis, it is possible to check the ADF Statistic. In this work, we refer to the ADF statistic as an ADF value. Usually, when this value reaches a positive representation it will describe a non-stationary time series. Otherwise, a negative value represents stationarity in a time series. The more negative this value is, the greater the confirmation that its statistic does not depend on the temporal offset.

### 2.3. Change Detection

Change detection is an algorithm that receives continuous data and maintains statistical information to be analyzed. It outputs an alarm if any change is detected.

**Definition** **3** (Change)**.**
*Change detection is usually applied to a data stream, which is an infinite sequence of data and can be represented by S, where S is given by: S = $\{\left(x_{1},y_{1}\right),\left(x_{2},y_{2}\right),\ldots ,$$\left(x_{t},y_{t}),\ldots \right\}$. Each instance is a pair $\left(x_{t},y_{t}\right)$ where $x_{t}$ is a d-dimensional vector arriving at the time stamp t and $y_{t}$ is the class label of $x_{t}$ [29]. In the case of univariate time series, the change detection algorithm will analyze each sample X of the time series at time t. It expects that at a given point $X_{t+n}$, where n is a time shift, has a distribution similar to the point $X_{t}$, otherwise an alarm will be triggered.*

Change detectors try to identify when a given change will occur by analyzing the behavior of previous data. Usually, change detectors are applied to data streams to identify concept drifts, which consist of changing data behavior over time [30]. This change is recurrent when performing data stream analysis [31]. The calibration process of these techniques depends on the choice of parameters that best fit the data behavior; techniques to improve their choice are often used. In this work, we seek to improve this choice with knowledge obtained from the time series stationarity. Considering that stationary series tend to have less variation over time, intervals that present less stationarity are more likely to present detection points for change detectors.

## 3. Related Work

Time series are explored in several types of applications and but not the subject of study until today. Most of these applications are focused on forecasting [32,33] and feature extraction [5] approaches, as can be seen in the following studies.

Due to the growth of generated data volume, there is a need for its treatment. Size reduction is an important procedure before carrying out analyses. The OSTS method was proposed by Carmona-Poyato et al. [22] and consists of the segmentation of points in a time series to reduce its size. This process is based on the A* algorithm and performs interpolations using optimal polygonal approximations. This method was used for comparison in this work with different segmentation techniques to assess its impact on the predictive capacity of new samples in a time series.

Time series segmentation from human specified breakpoint detection was introduced in Lee et al. [6]. The process is conducted by exploiting deep learning techniques for automatically extracting features. However, this technique was not used as a preprocessing step to predict new samples in a time series. Despite it being very interesting, it requires much data processing before applying breakpoint detection. For our approach, we explored an automatic way of detecting these breakpoints simply from the analysis of time series stationarity.

Temporal segmentation is also an alternative as discussed by Bessec et al. [23] and Jamali et al. [20]. The former performs predictions at a day-ahead with combinations of linear regression techniques, Markov-switching models, and thresholds. The latter requires user knowledge to define the ideal parameters for the proposed segmentation method, based on the time series characteristics. Both works rely on prior knowledge and since they are applied in specific environments, it is difficult to generalize. The work addressed by Hooi et al. [34] also performs time series segmentation, but not in a temporal approach, it is based on patterns to create a vocabulary capable of identifying possible regions to perform segmentation. It is dependent on user intervention and knowledge.

Other proposals are derived from classical methods, such as the survey by Keogh et al. [3], which describes traditional representativeness techniques from the perspective of data mining. Different segmentation methods and the evaluation of their impact on representativeness in the time series are discussed; most of them are based on linear approximations. Among the methods used for comparison, there are sliding windows, bottom-up, and top-down approaches. There is also a proposal by the authors called SWAB, which consists of the union of the sliding windows and Bottom-up methods. This study shows a classic approach to segmentation in time series, where the samples to be evaluated do not depend on static information inherent in their behavior. In this work, techniques placed on stationarity are proposed to perform segmentation, as an alternative method to improve the predictive capacity of new samples.

Table 1 presents a summary of some relevant works in the literature that address the topic of time series segmentation. To our knowledge, our segmentation proposal differs from the others in Table 1 because it is the first to deal with segmentation to improve new sample predictions through an approach based on time series stationarity.

## 4. Proposed Approach

The proposed approach to perform the time series segmentation can be examined in Figure 1 and  Figure 2. They demonstrate the steps followed to identify potential samples that do not contribute to the time series stationarity. A framework and technique to perform the segmentation of a time series are proposed. In both cases, the process begins with the stationarity analysis of a given time series, defined as the ADF test. Each segmentation proposal has its peculiarities, but both depend on change detector processes to define the samples that should be segmented. The main objective of sample removal is to obtain a better-behaved time series model, which can improve various applications, such as prediction, data compression, and feature extraction. The framework, called Change Detector segmentation, seeks to verify the impact of stationary analysis in aiding change detector techniques. Intervals with less stationarity serve as an extra alert for the change detector techniques to perform their detection. The ADF-based segmentation consists of a stationary analysis of the time series. The framework was created because as far as we know, there are no stationarity-based time series segmenters in the literature. Another reason is that choosing hyperparameters for these techniques is not a trivial task. The difference between the framework and ADF-based segmentation is that the latter does not rely on the windowing process to perform segmentation. The process is carried out continuously.

### 4.1. Change Detector Segmentation Framework

The Change Detector Segmentation is a framework to perform segmentation in time series. Segmentation is based on the analysis of critical intervals in the time series alongside change detectors and stationarity analyses. Algorithm 1 describes how the segmentation points are obtained.
**Algorithm 1: **Change Detector Segmentation code
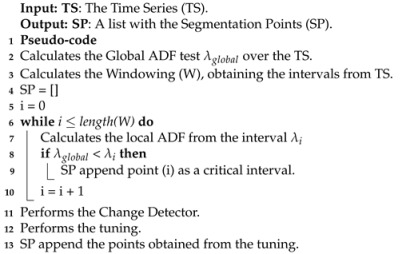


Figure 1 demonstrates the entire process for performing the segmentation, each step will be defined below:

#### 4.1.1. Global ADF Test

The Global ADF test obtains the stationarity value of all time series samples, and it is applied according to the definition in Section 2.2. It is denoted by $\lambda_{global}$. This step is done to obtain information about the general behavior of the time series. The ADF value is obtained as the output, which serves as a threshold for evaluating each point in the time series. This process can be replaced by another one that is also able to describe the behavior of the time series.

#### 4.1.2. Windowing

The windowing step consists of checking cyclical behaviors in a given time series to define its size. It can be given for days, weeks, years, or even longer intervals depending on the time series. The windowing allows comparing the Local ADF test of each interval with $\lambda_{global}$, described in the Thresholding box in Figure 1. This process is conducted to identify critical intervals, which consists of $\lambda_{i}$ greater than $\lambda_{global}$ and makes the series less stationary. The critical intervals will be evaluated by the change detectors to examine samples individually, as they have more variability in the time series.

#### 4.1.3. Local ADF Test

The Local ADF test obtains the stationarity value of a given interval of the time series and is denoted by $\lambda_{i}$, where *i* represents an interval at a given time. This interval is obtained by the windowing step and will be inputted into the thresholding step.

#### 4.1.4. Thresholding

The Thresholding step compares the stationarity value of a given interval $\lambda_{i}$ with $\lambda_{global}$ for decision making. This interval is considered critical when $\lambda_{i}$ presents a value greater than $\lambda_{global}$. This configuration implies that this critical interval does not present a stationary behavior like the remaining samples in the time series. A possible problem at this stage is to consider every sample inside the critical interval to be segmented. However, it is mitigated by the tuning step.

#### 4.1.5. Tuning

A tuning process was designed to make the change detectors individually examine each sample with greater emphasis on critical intervals. In addition to the use of standard hyperparameter values for the change detectors, an attempt to improve their choice was also made. The standard values were varied to fit the best configuration for each scenario. Three different approaches were considered for tuning to optimize the choice of the best hyperparameters, The Bigger in Smaller out, Bigger in, and Smaller out approaches. These approaches allow obtaining three different hyperparameter variation results according to the available critical intervals.

**Bigger in Smaller out:** this strategy chooses the hyperparameter configuration that keeps the largest amount of data within the critical intervals and the least amount of data outside these intervals. This approach balances the selection of samples present in the critical interval with the least number of samples outside it.**Bigger in:** this strategy chooses the hyperparameter configuration that keeps the largest amount of data that is within the critical interval. This approach places the greatest emphasis on removing samples at critical intervals.**Smaller out:** this strategy chooses the hyperparameter configuration that keeps the least amount of data outside the critical interval. This approach minimizes the selection of samples outside the critical intervals.

#### 4.1.6. Change Detectors

Change detectors are usually applied to data streams; however, it is also possible to extend them to identify changes in time series [35]. As its name suggests, it seeks to identify changes in behavior in a data stream. This step is applied to identify the points of the time series that do not present stationary behavior. As the output, it generates a set of points to perform the removal.

#### 4.1.7. Removing Samples

This step consists of removing the samples from the time series according to the points provided by the change detectors. This is the last step of the proposed framework. Neighbor points that were not removed from the time series are interpolated.

### 4.2. ADF-Based Segmentation

The ADF detector is also a proposal for this work. It performs the segmentation of samples in the time series based on their stationarity. Its process starts with the Grace Period, which consists of calibrating the ADF test with a predetermined interval of the time series, according to Figure 2. It consists of separating a small number of time series samples to calibrate the ADF test, e.g., 10% of the initial time series samples. After this calibration, the Local ADF test is obtained. From this point on, the remaining samples will be evaluated to verify their stationarity continuously. For each new sample, a new ADF test value is acquired. If this value is less than the current Local ADF test value, we understand that this sample must be kept in the time series by maintaining stationary, and the Local ADF test value is updated. However, if the sample has a greater value of ADF test than the Local ADF, it is assumed that this sample should be removed. This comparison between these two values depends on an α factor, where an acceptable error rate is applied. In this work, the values 0.05, 0.1, and 0.15 were used for α. This process seeks to ensure that the stationarity of the time series is not affected by samples that are out of normal behavior. Moreover, it can reduce the time series size. Algorithm 2 describes how the segmentation points are obtained.
**Algorithm 2: **ADF-based Segmentation code
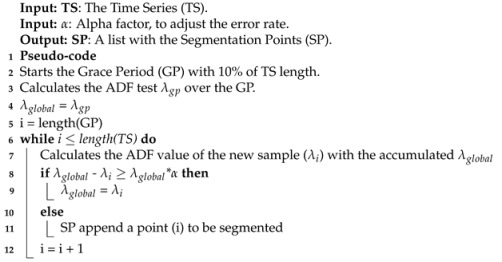


## 5. Experimental Study

The experimental study aims to compare the results obtained with works in the state of the art based on time series segmentation to answer the following Research Questions (RQ).

RQ1: Can time series segmentation based on stationarity analysis assist in the predictive process of new samples?RQ2: Is stationarity-based segmentation capable of providing improvements in the aspect of prediction and reduction in time series sizes in different databases?RQ3: Were the segmentation techniques proposed in this work compared with similar techniques?RQ4: How correlated are stationarity and segmentation techniques?

### 5.1. Experimental Setup

#### 5.1.1. Databases

The databases containing the time series used in this work are divided into three groups. The first group consists of a real database obtained by a Photovoltaic Plant (PV) installed at the State University of Londrina, Brazil. This database presents samples of power generation that started to operate in November 2019. The plant has 1020 solar panels. The samples are collected every 15 min. Thus, 96 samples are collected each day. To obtain better efficiency in capturing solar energy, the plant usually operates from 6 a.m. to 7 p.m. The sensors of the photovoltaic plant are capable of capturing 18 different features provided by solar energy. Among them, the generated power is used for analysis in this work.

The second group consists of the Minimum Daily Temperatures (MDT) database, which has the minimum daily temperatures over 10 years (1981–1990) in the city of Melbourne, Australia [36].

The third group consists of the database known as the Monthly Sunspot (MS) database [37]. This database describes a monthly count of the number of observed sunspots over 230 years (1749–1983).

The experiments were separated at different intervals to verify different variations in the time series. For the first group, represented by the PV database, we have intervals starting from November, since it is the initial month of data collection and is subject to greater variation.

The experiments were separated into different training and testing intervals to verify unusual variations in the time series. For the first group, represented by the PV database, we have intervals starting from November, since it is the initial month of data collection and is subject to greater variation. For the second group, three intervals were selected. The first two intervals consist of a division of the third interval. Finally, the third group is comped by an interval of 150 years for training and 83 years for testing. The selection of these intervals was made without specific criteria. The objective is to verify the improvement in predictive capacity through the segmentation process proposed in this work.

Table 2 presents information from the databases used for experimentation with some additional information. Each database was identified by a letter from A to J for better identification in the results section. The ADF value column represents the stationarity of each training interval. These values demonstrate that the PV base presents well-behaved samples compared to other databases, since it has a smaller ADF value. Furthermore, it is possible to verify the relationship of these values with the results obtained in the reduction of dataset size and predictive capacity. The exact intervals used for experimentation for each database and further information can be verified in Table 2.

After determining each experiment interval, the time series were submitted to segmentation techniques according to the ones proposed in this work, and their original versions were subjected to prediction techniques. This step consists of verifying whether the segmentation process improves the predictive performance of a time series, besides decreasing its size.

#### 5.1.2. Change Detectors

The change detectors used in this work were ADW [38] and PH [39], present in the scikit-multiflow package [40]. They were used as detectors in the change detector step of the framework to compare the efficiency and interference of a particular detector algorithm. The choices for these detectors were due to their adaptability to time series. ADW is an adaptive sliding window algorithm for detecting changes and has a delta parameter to perform its detection. The PH detector works by computing the observed values and their mean up to the current moment [39]. It is composed of four parameters to perform its detection: the minimum number of instances before detecting change, the delta factor for the PH test, a threshold called lambda factor, and the forgetting factor, to weight the observed value and the mean.

The hyperparameters of both detectors were varied with values in percentage to values below and above the standard ones to obtain a configuration that best fits the proposed segmentation of the time series samples. For experimentation in this work, increments of 25% were used, starting at half of the standard value until double its standard value.

#### 5.1.3. Metrics

The metric adopted for predictive gain assessment was the Root Mean Squared Error (*RMSE*), which is given by the following equation:(4)$$RMSE=\sqrt{\frac{1}{n}\Sigma_{i=1}^{n}{\left(\frac{{\hat{y}}_{i}-y_{i}}{n}\right)}^{2}}$$
where ${\hat{y}}_{i}$ represents the predicted sample and $y_{i}$ represents the actual value of the sample at time *i*.

#### 5.1.4. Prediction Techniques

Four predictive techniques are used in this work. The Naive, ARIMA, LSTM, and TCN methods. All of these predictors are used over the segmented and original time series.

The first predictive technique consists of a benchmark approach for time series, due to its simplicity and considerable acceptance of its results. Its operation considers that the next value to be obtained will be equal to the last observed value. Equation (Equation 5) shows how these values are obtained:(5)$${\hat{x}}_{t+1}=x_{t}$$
where ${\hat{x}}_{t+1}$ represents the next value to be predicted and $x_{t}$ represents the last observed value.

ARIMA is essentially exploratory and seeks to fit a model to adapt to the data structure [24]. With the aid of the autocorrelation and partial autocorrelation functions, it is possible to obtain the essence of the time series so that it can be modeled. Then, information such as trends, variations, cyclical components, and even patterns present in the time series can also be obtained [41]. This allows the description of its current pattern and prediction of future series values [42]. Autoregressive models have been used for a long time in time series analysis and are therefore considered popular and simple. However, some studies (Akhter et al. [43]; Cerqueira et al. [44]) demonstrate that when dealing with a very large amount of data, models based on DL tend to have better results.

As a comparison, the LSTM and TCN techniques, which are based on DL, are evaluated in this work as a comparison to the classical ARIMA and Naive models. LSTM is a type of RNN (Recurrent Neural Network). Unlike some traditional neural networks, LSTM can remember the most useful information. This is possible thanks to its architecture. The networks that comprise the LSTM are connected in the form of loops. This process allows information to persist on the network. It also has a gating mechanism for learning long-term dependencies without losing short-term capability [17]. This mechanism allows for what we call a neural network that has memory. This feature is essential when dealing with time series, as some moments tend to be repeated over an interval of time. Another important feature is the stationary approach of this work, which allows the neural network model to adapt faster to the behavior of the time series. However, these approaches based on deep models are only interesting for problems that have a large amount of data, due to their complexity. For simpler scenarios, statistical models such as ARIMA tend to fill the need.

TCN is a special type of Convolutional Neural Network capable of handling a large amount of information [25]. This processing is done through causal convolution, which ensures the model cannot violate the order in which the data is processed. TCN uses a one-dimensional fully convolutional network architecture, where each hidden layer has the same length as the input layer, and zero padding of length [45]. As highlighted for LSTM, this approach tends to perform better for more complex problems where a statistical model is not able to determine long correspondences between the data.

Table 3 presents the hyperparameters used for experimentation. The Naive technique does not have an extra configuration as it is based only on the value of the previous sample. For the ARIMA modeling technique auto-arima from pmdarima( https://pypi.org/project/pmdarima/ assessed on 2 August 2021) library was used. As for the LSTM and TCN, the values in Table 3 were used to obtain the best possible configuration for each of them.

### 5.2. OSTS Method

To answer RQ3, we compare our proposed techniques to the OSTS. The OSTS technique was proposed by Carmona-Poyato et al. [22] and consists of an approach to reduce dimensionality in a time series. In essence, it can locate the maximum and minimum points in a time series. This process is done through numerical approximation considering a point $P_{j}$ and analysis of neighboring points. When the sample $P_{j}$ is removed, its neighboring points are joined in a straight line. Even though its segmentation approach has no motivation for predictive improvement, it was used for comparison with the proposed techniques in this work. Its choice is also due to the extensive comparison of the OSTS approach with other segmenters. It also differs from classical approaches based on sliding windows and thresholds of statistical moments [3].

### 5.3. Experimental Results

In this subsection, the results obtained by the experiments will be presented. Table 4 and Table 5 present the prediction results in terms of RMSE, demonstrating the predictive gain through time series segmentation. These results are established on the test interval defined in Table 2 using the four prediction techniques defined in the Section 5.1.4. The ADW and PH segmenters present the results of the Change Detector Segmentation Framework. These results are based on the best variation of the tuning process, defined in the Section 4.1.5. The ADF-based segmentation is represented by ADF. Only the choice of the α factor that obtained the best result was considered.

#### 5.3.1. Predictions Results

To answer RQ1, we have the predictive results applied to the datasets defined in this work with the proposed segmentation techniques. Table 4 presents the global error results of the new sample prediction from all databases, A to J, while Table 5 shows the relative error between them. In Table 4, we have highlighted in the last line the accumulated predictive error of all original databases, without the segmentation process. The OSTS segmenter was not able to improve the predictive capacity of the predictors in any case, except for the Naive predictor. It had the highest RMSE among the other segmentation techniques. For the PH segmenter, it obtained the second highest RMSE and did not show predictive improvement, except for the Naive predictor. As for the ADW and ADF segmenters, there were predictive improvements. In the case of ADW, only the TCN predictor was not able to present a predictive improvement; however, it obtained the smallest RMSE for the ARIMA and LSTM predictors. For the ADF segmenter, only ARIMA had no improvement; however, the smallest RMSE was obtained for the Naive and TCN predictors.

For Table 5, we have the relative RMSE between the predictive performance of the original database with the proposed segmentation techniques. Each database, A to J, is made up of four different prediction techniques. The segmentation techniques are presented in a decreasing ranking from the best result to the worst. In summary, the results that obtained a relative RMSE above one do not present a significant predictive performance. It is also possible to analyze why the original distribution did not obtain better predictive performance in any of the bases for the ARIMA and TCN predictors. Validating the results obtained by Table 4, it is verified once again that the most significant predictive results are between the LSTM and TCN predictors, promoted by the ADF and ADW segmenters.

In general, it can be said that the proposed segmentation techniques performed well in all experienced databases. Only in 6 cases out of 40 did the segmentation techniques not provide a better predictive capacity. Of these six, three are located in the Naive predictor, which has the worst predictive power among the other predictors. Moreover, even though it did not obtain a better predictive performance, its result was the same as the original distribution, but with the advantage of decreasing the time series size.

#### 5.3.2. Time Series Size Reduction

In this subsection, the effect of the segmentation process in reducing the time series size is discussed. RQ2 is answered with the analysis presented in Table 4 and Table 5, in the previous subsection, and the results in Figure 3, since it presents a comparison of the size reduction of the segmentation techniques for each database. These reductions are based on the LSTM predictor since it achieved the best predictive performance.

We can affirm through the analysis of Figure 3 that the size reduction of the time series is not directly related to the predictive gain. The OSTS segmentation technique was the one that obtained the biggest size reduction, but the worst result of the predictive gain, according to Table 4 and Table 5. We can also affirm that the size reduction is not so considerable when comparing the other segmentation techniques, but it is still able to improve the performance gain. The ADF, ADW, and PH detectors achieved similar reductions in the PV databases, while the PH detector showed greater variability in the other databases. In addition, considering that these results were obtained using the LSTM predictor, it is possible to conclude that the ADF and ADW detectors can improve the predictive gain of new samples, while reducing the time series size.

An interesting link that can be made is that the databases that had the biggest reductions were the ones with the highest ADF values in Table 2. The same is not verified in the predictive improvement (Table 5), since the cases that obtained better results are concentrated in the more stationary databases, e.g., database C.

#### 5.3.3. Sample Segmentation

Figure 4 and Figure 5 present sample segmentation intervals in two distinct databases, A and C, respectively. In Figure 4, it is possible to observe in the sample interval 500 to 515, none of the techniques presented a region to be segmented, it being considered an essential region to describe a stationary behavior. Near sample 545 a peak is seen, which was highlighted by most detectors as segmentable. The OSTS and ADW detectors acted at the moment and just before the appearance of this point, while the ADF acted at the moment of descent from this peak. The PH detector did not act to identify this region. However, the excess of points identified by the OSTS detector and the absence of others in the evaluated interval demonstrates great disparity, as the other detectors pointed out this highlighted interval with a high level of stationarity, requiring few segmentations.

Figure 5 exhibits a behavior similar to the database analyzed above, but with emphasis on the samples verified from point 562, where there were many segmentation points by ADF. This demonstrates that the ADF detector operates differently from ADW and PH. This interval highlighted by ADF demonstrates an important characteristic when these samples were detected and proved to be more sensitive for stationarity analysis.

### 5.4. Statistical Analysis

Friedman’s statistical test and the post-hoc test of Nemenyi were used to verify the statistically significant difference between the performance of the segmentation techniques proposed in this work. This test was performed for each predictor described in this work and can be evaluated by Figure 6, Figure 7, Figure 8 and Figure 9. The critical difference (CD) allows checking when there were statistical differences between the segmenters, each diagram and algorithms’ average ranks are placed in the horizontal axis, with the best ranked to the right. For ADW segmenter, it is possible to observe that there were static differences compared to the original time series in all scenarios. A black line connects algorithms for which there is no significant performance difference.

Figure 6 demonstrates no statistical difference between the ADF and OSTS techniques compared to the original time series. There was also no statistical difference between OSTS and ADW. For the Naive predictor, only the ADW and PH techniques showed statistical differences from the normal distribution.

For the ARIMA predictor in Figure 7, the OSTS technique showed the worst performance. The PH and ADF techniques did not present statistical differences with the original distribution, only the ADW reached differences in this case. Finally, for the LSTSM and TCN predictors shown in Figure 8 and Figure 9, respectively, there were no statistical differences for the OSTS and PH techniques compared to the original distribution. In both cases, the ADF and ADW techniques showed better performance, with statistical similarity.

It is worth mentioning that segmentation techniques based on the analysis of stationarity of a time series are a viable technique. The advantages of this approach are the simplicity in how each sample is evaluated and the independence of information and feature extraction to perform the segmentation. In addition, we consider this as a novel approach since the segmentation of time series to improve the predictive capacity of new samples is not a task found in the literature.

Another consideration is that the segmentation process must be able to maintain the representativeness of the time series and our approach guarantees this. A large reduction in the time series size does not guarantee better performance, as seen in the results of the OSTS detector.

### 5.5. Stationarity Impact on Segmentation

One of the pillars of our hypothesis for creating a framework proposal and segmentation method is stationarity analysis. Thus, RQ4 is answered in this subsection. Extracting two stationarity descriptors based on the ADF test, in particular, the standard deviation of stationarity (Std Stationarity) and the mean value of stationarity (Mean Stationarity), we computed their correlation using the Pearson Correlation test. This test is related to the errors (RMSE) obtained from the segmentation techniques proposed in this work and the OSTS method. Figure 10 represents, as a heatmap, the correlation between stationarity and RMSE of all segmenters.

As observed in Figure 10, OSTS RMSE does not correlate with the stationarity of the time series, reaching 0.01 and −0.02 of correlation, standard deviation, and mean. This is an expected phenomenon, since the strategy of OSTS is driven by the neighborhood of maximum and minimum local values. On the other hand, ADF, ADW, and PH presented an important correlation with the standard deviation of stationarity. More precisely, they obtained a positive correlation, which reveals that the increase in stationarity variation in the time series leads to a less effective segmentation. However, high mean stationarity leads to boosted results from segmentation, with PH obtaining the highest correlation.

Considering the perspective of dimension reduction, we evaluated the correlation between reduction rate and stationarity, as previously, for performance. OSTS reduction capacity is inversely correlated with the stationarity mean, −0.98. Regarding the change-based approaches, PH reduction capacity was the best correlated with an inverse correlation of −0.86, as shown in Figure 11. For ADF and ADW, we have the lowest correlation between reduction rate and stationarity, −0.29 and −0.38, respectively.

Overall, these results indicate that the stationarity of time series is correlated with the dimension reduction capacity provided by the methods. It is worth mentioning that ADW and PH followed the same framework and tuning strategy, but the way stationarity correlates to their reduction capacity and predictive performance diverges. This fact emphasizes the need for caution when selecting the change detector according to each time series pattern.

It is also worth noting that our stationary strategy to perform segmentation maintains the original time series behavior and removes only disposable samples, which makes it a less invasive method.

Finally, the standard deviation of stationarity exposed how tricky the segmentation might be when reducing the predictive error of high-variated stationarity patterns.

### 5.6. Limitations

A limitation identified in our proposal is the need to have prior knowledge about the time series cyclical behaviors. This information ensures better performance in identifying less stationary samples.

Another possible limitation is the grace period phase that performs the ADF-based segmentation calibration process. It is another prior information that guarantees better performance in identifying less stationary samples.

## 6. Conclusions

The stationary approach for segmenting samples in time series has a positive impact on the issue of prediction of new samples and size reduction. According to the results, it is possible to affirm that the segmentation techniques had a better performance for the prediction algorithms based on DL. The ADW and ADF segmentation had the best performance in the DL approaches, with ADW for LSTM and ADF for TCN. However, an advantage over the functioning of the ADF over the ADW is the independence of a priori knowledge about the time series. The ADF segmentation analysis demands only a small number of samples to perform its grace period and start the segmentation process and has only one hyperparameter.

The largest cases of time series size reduction came from the OSTS technique, achieving 0.40 reduction compared to the original distribution, but this result does not follow the predictive performance of the new samples. A balance between the reduction and prediction approach is best seen by the ADF and ADW segmenters.

Stationary analysis proved to be a great ally in the study of time series behavior and more studies can be directed in this respect beyond sample segmentation. For future work, we seek to improve segmentation techniques aimed at the data stream, where the segmentation would be given in real-time, as the data is obtained. Furthermore, improving the extraction of a priori information from the time series, makes these methods more autonomous.

## Figures and Tables

**Figure 1 sensors-21-07333-f001:**
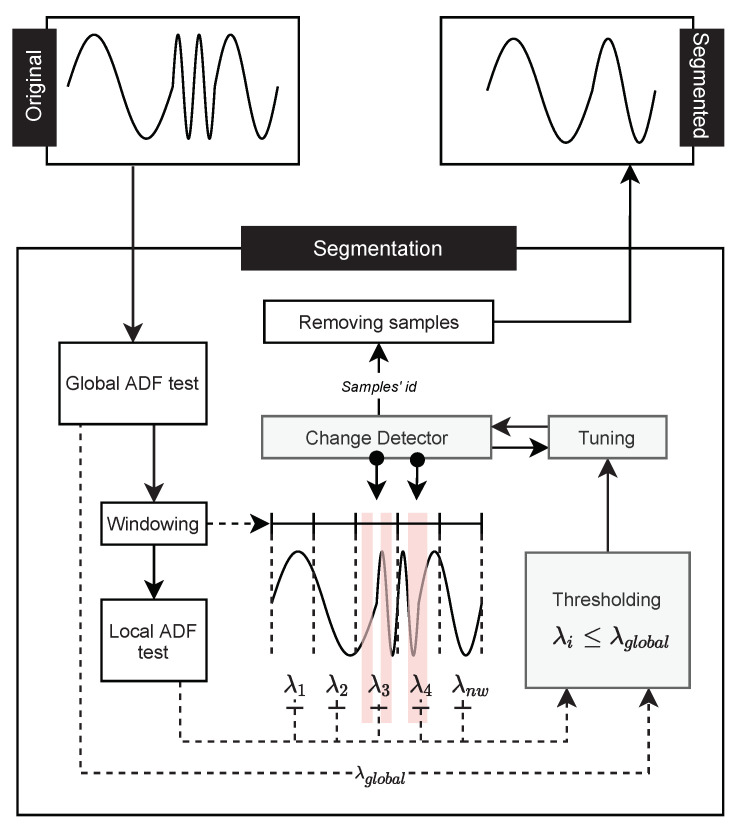
Methodology framework of Change Detector Segmentation.

**Figure 2 sensors-21-07333-f002:**
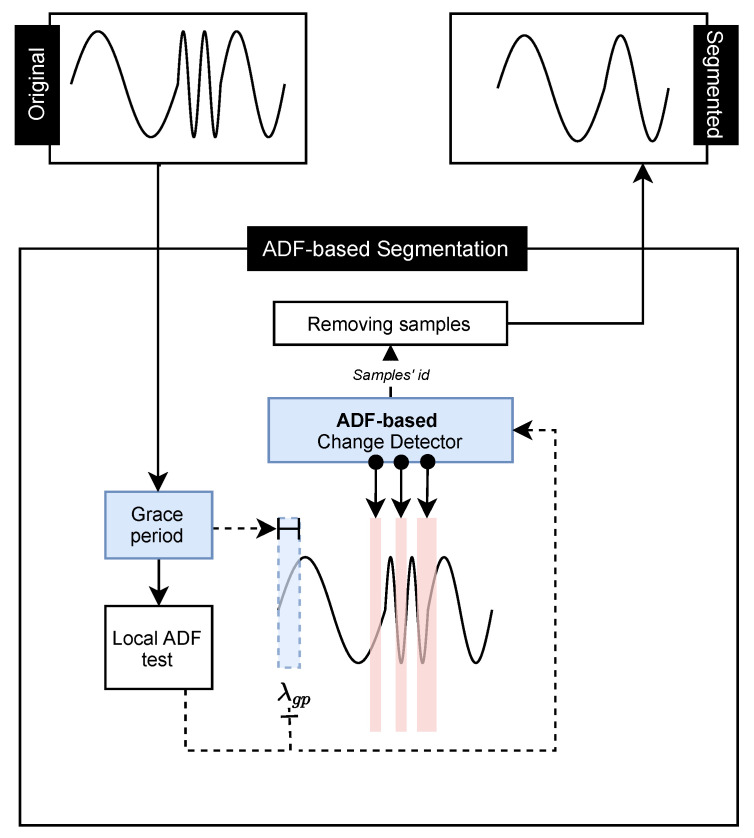
Methodology proposal of ADF-based Segmentation.

**Figure 3 sensors-21-07333-f003:**
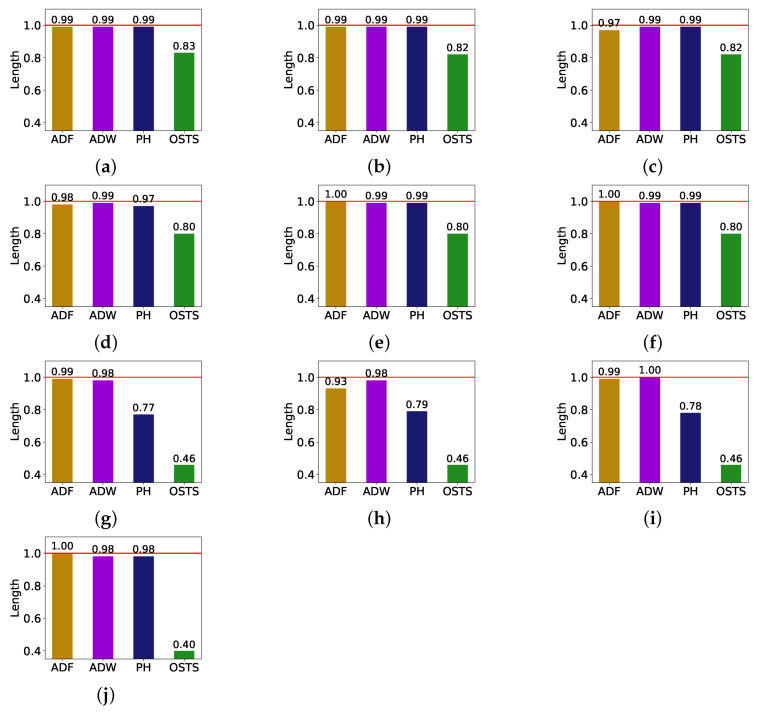
Time series size reduction according to segmentation methods ADF, ADW, PH, and OSTS. (**a**) Database A; (**b**) database B; (**c**) database C; (**d**) database D; (**e**) database E; (**f**) database F; (**g**) database G; (**h**) database H; (**i**) database I; (**j**) database J.

**Figure 4 sensors-21-07333-f004:**
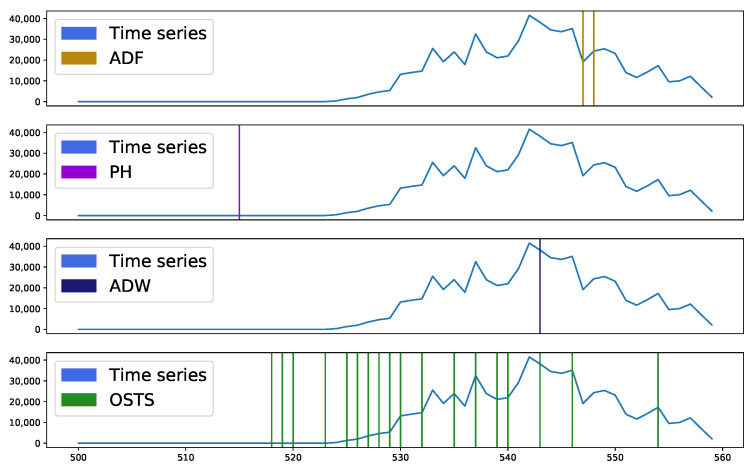
Segmented samples of dataset A (chunk from 500 to 560 samples). ADF removed 2 samples, PH and ADW removed a single sample, and OSTS removed 17. The stationarity-based segmenters considered the range from 515 to 543 as stable and without the need for segmentation, while the OSTS segmenter performed quite a lot of segmentation at these points.

**Figure 5 sensors-21-07333-f005:**
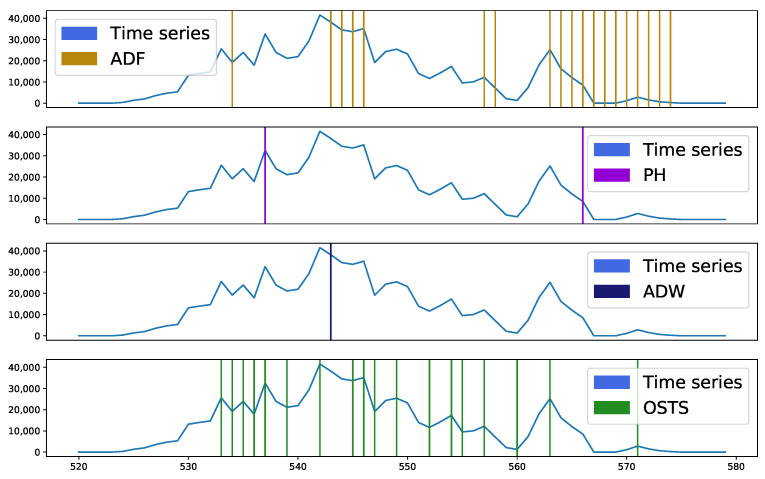
Segmented samples of dataset C (chunk from 520 to 580 samples). ADF removed 19 samples, PH removed 2 samples, ADW removed a single sample and OSTS removed 18. Unlike the case of dataset A, the ADF segmenter performed many segmentations like the OSTS, but in different regions, while the other segmenters considered the region as stable.

**Figure 6 sensors-21-07333-f006:**
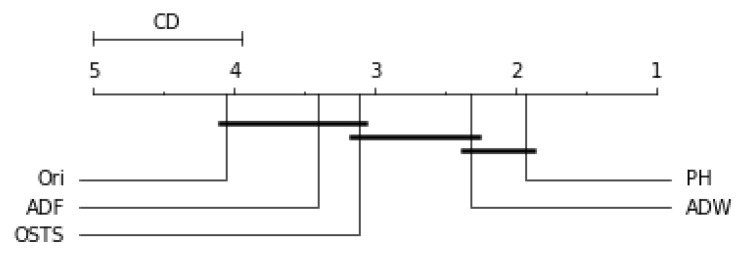
Comparison of the RMSE values obtained by segmentation techniques for Naive predictor according to the Nemenyi test. Groups that are not significantly different (α=0.05 and CD=1.04) are connected.

**Figure 7 sensors-21-07333-f007:**
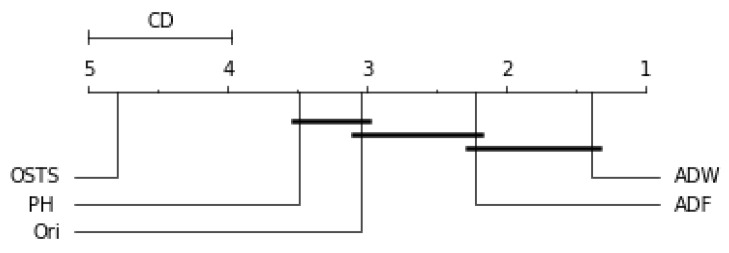
Comparison of the RMSE values obtained by segmentation techniques for ARIMA predictor according to the Nemenyi test. Groups that are not significantly different (α=0.05 and CD=1.03) are connected.

**Figure 8 sensors-21-07333-f008:**
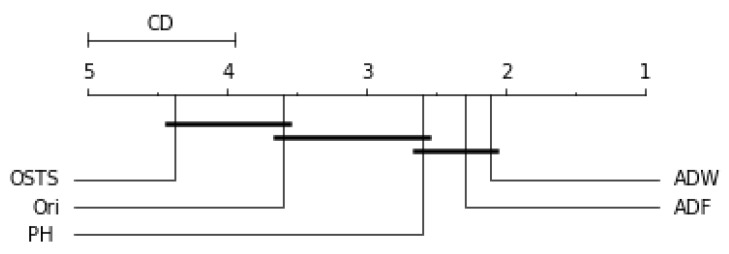
Comparison of the RMSE values obtained by segmentation techniques for LSTM predictor according to the Nemenyi test. Groups that are not significantly different (α=0.05 and CD=1.04) are connected.

**Figure 9 sensors-21-07333-f009:**
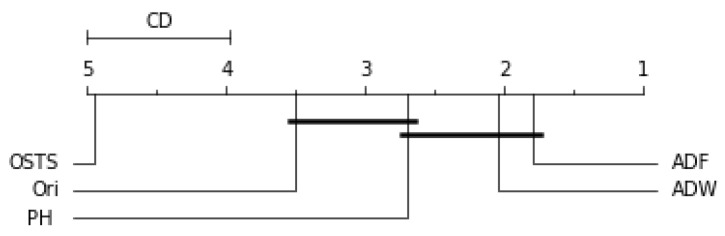
Comparison of the RMSE values obtained by segmentation techniques for TCN predictor according to the Nemenyi test. Groups that are not significantly different (α=0.05 and CD=1.03) are connected.

**Figure 10 sensors-21-07333-f010:**
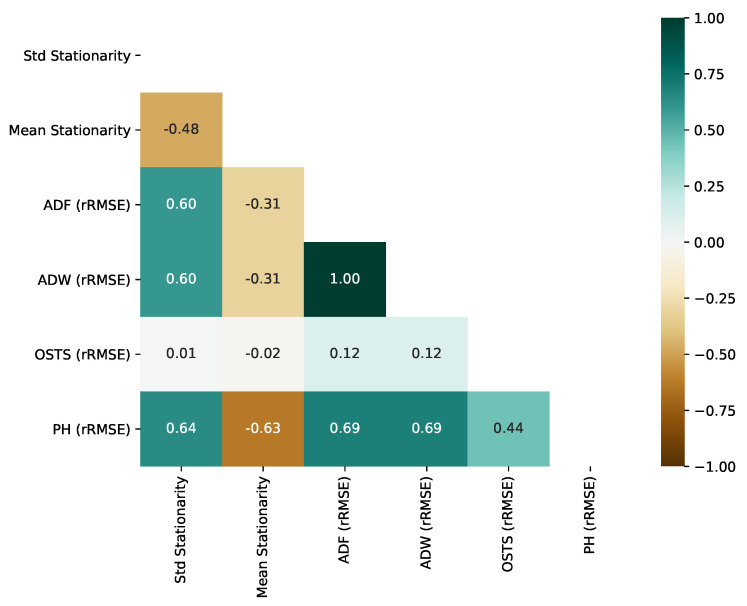
Correlation between the stationarity (based on ADF Test) and performance (rRMSE) obtained by segmentation methods.

**Figure 11 sensors-21-07333-f011:**
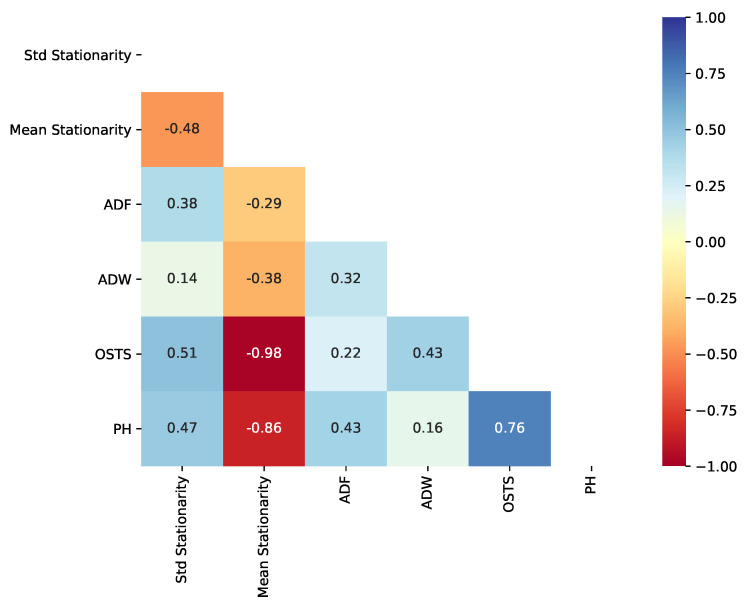
Correlation between the stationarity (based on ADF Test) and dimension reduction delivered by segmentation methods.

**Table 1 sensors-21-07333-t001:** Comparison of time series segmentation techniques in the literature.

Reference	Segmentation Technique	Purpose of Segmentation
Carmona-Poyato et al. (2020) [22]	Based on A* algorithm with optimal polygonal approximations	Data representation reducing the dimensionality with minimum information loss
Lee et al. (2018) [6]	Unsupervised approach, based on deep learning	Automatic knowledge extraction
Hooi et al. (2017) [34]	BeatLex, based on patterns to match segments of the time series	Vocabulary-based approach to match segments of the time series in an intuitive and intepretable way
Bessec et al. (2015) [23]	Temporal segmentation based on hourly and seasonal segmentation	Forecast spot prices in France with double temporal segmentation
Jamali et al. (2015) [20]	Temporal segmentation based on thresholds of the time series features	Segments the changes in the vegetation time series to identify the change type and its characteristics
Keogh et al. (2004) [3]	Sliding windows, Bottom-up, Top-Down, and SWAB	Empirical comparison of time series segmentation algorithms form a data mining perspective

**Table 2 sensors-21-07333-t002:** Intervals used for experimentation from PV, MDT, and MS databases.

Identifier	Database	Train Interval	ADF Value	Test Interval
A	PV	November to December	−16.57	4 weeks January
B	PV	November to January	−19.24	March
C	PV	November to February	−21.81	March
D	PV	January to February	−16.36	4 weeks March
E	PV	February	−11.57	4 weeks March
F	PV	February	−11.57	days in March
G	MDT	1981 to 1984	−3.14	1985
H	MDT	1986 to 1989	−2.59	1990
I	MDT	1981 to 1989	−4.34	1990
J	MS	1749 to 1899	−7.04	1900 to 1983

**Table 3 sensors-21-07333-t003:** Hyperparameters experimented with for tuning.

	Parameters	Experimented Hyperparameters
LSTM	Number of stacked layers	1, 2, 3
	Units	32, 64, 128
	Dropout	0
TCN	Number of filters	32, 64
	Kernel	2, 3
	Dilations	[1, 4, 12, 48], [1, 2, 4, 8, 12, 24, 48], [1, 4, 16, 32], [1, 2, 4, 8, 16, 32], [1, 3, 6, 12, 24], [1, 2, 6, 12, 24], [1, 2, 4, 8, 16], [1, 4, 16], [1, 2, 4, 8], [1, 4, 8]
	Blocks	1, 2
	Dropout	0

**Table 4 sensors-21-07333-t004:** Global RMSE results from predictive techniques.

	Naive	ARIMA	LSTM	TCN
**ADF**	**18,923.65**	13,559.87	3193.70	**3242.92**
**ADW**	18,923.66	**13,532.41**	**3177.46**	3294.75
**PH**	**18,923.65**	13,727.95	3408.02	3844.67
**OSTS**	**18,923.65**	14,239.36	3482.60	3952.74
**Original**	20,006.11	13,537.78	3229.33	3276.01

**Table 5 sensors-21-07333-t005:** Relative RMSE of each segmentation technique referring to the original base across four different predictive techniques using the ADF-based method, ADW, and PH (i.e., instances of our segmentation framework), and OSTS segmentation method. Lower errors are in bold and worst average cases are underlined.

Database Identifier	Segmentation Techniques	Predictive Techniques				Average
		Naive	ARIMA	LSTM	TCN	
A	ADF	**0.99**	**0.99**	**0.97**	**0.99**	**0.98**
	ADW	**0.99**	**0.99**	1.00	**0.99**	0.99
	PH	**0.99**	**0.99**	1.03	**0.99**	1.00
	OSTS	1.00	1.06	1.01	**0.99**	1.01
B	ADF	**0.99**	1.00	**0.97**	0.99	**0.98**
	ADW	1.00	**0.99**	0.99	**0.98**	0.99
	PH	**0.99**	**0.99**	0.98	0.99	**0.98**
	OSTS	1.08	1.12	1.00	1.02	1.05
C	ADF	1.00	**0.98**	**0.94**	0.83	**0.93**
	ADW	1.00	1.02	0.96	0.85	0.95
	PH	1.00	1.02	0.96	**0.81**	0.94
	OSTS	**0.98**	1.03	0.99	1.10	1.02
D	ADF	1.00	**0.95**	1.02	0.99	0.99
	ADW	1.00	0.98	1.03	0.97	0.99
	PH	**0.99**	0.98	1.01	**0.90**	**0.97**
	OSTS	1.00	1.00	1.34	1.30	1.16
E	ADF	**0.99**	**0.99**	1.00	0.98	**0.99**
	ADW	1.00	**0.99**	1.01	0.99	**0.99**
	PH	**0.99**	**0.99**	1.02	**0.96**	**0.99**
	OSTS	1.00	1.01	1.26	1.34	1.15
F	ADF	1.00	1.00	0.87	**0.73**	0.90
	ADW	**0.99**	**0.99**	**0.67**	0.86	**0.88**
	PH	**0.99**	1.00	0.74	1.20	0.98
	OSTS	**0.99**	1.03	0.93	1.45	1.10
G	ADF	1.00	1.00	**0.99**	0.96	0.98
	ADW	1.00	**0.99**	**0.99**	**0.92**	**0.97**
	PH	1.00	1.00	**0.99**	0.95	0.98
	OSTS	1.00	1.22	1.05	1.17	1.11
H	ADF	1.00	**0.87**	**0.95**	0.99	**0.95**
	ADW	1.00	0.99	**0.95**	0.99	0.98
	PH	1.00	0.93	0.96	**0.96**	0.96
	OSTS	1.00	1.24	1.00	1.12	1.11
I	ADF	1.00	1.00	**0.99**	**0.98**	**0.99**
	ADW	1.00	**0.99**	1.00	1.00	**0.99**
	PH	1.00	**0.99**	**0.99**	0.99	**0.99**
	OSTS	1.00	**0.99**	1.00	1.04	1.00
J	ADF	**0.99**	**0.98**	1.00	0.98	**0.98**
	ADW	**0.99**	**0.98**	1.00	**0.97**	**0.98**
	PH	**0.99**	1.00	1.00	1.00	0.99
	OSTS	**0.99**	0.99	1.22	1.13	1.08

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
