# Peer review of "Time Series Segmentation Based on Stationarity Analysis to Improve New Samples Prediction"

_sensors, 2021, doi:10.3390/s21217333_

Round 1

Reviewer 1 Report

This interesting paper proposes new time series segmentation methods based on stationarity analysis by means of the augmented Dickey-Fuller test. The effectiveness of the proposed methods is demonstrated by performing realistic time-series data. The relationship between the proposed methods and the switching state-space model should be discussed in detail. 

Reviewer 2 Report

The paper presents a new proposal for time series segmentation based on stationarity, denominate ADF Based Segmentation. By a framework to perform segmentation of time series based on stationarity using change detector algorithms, called Change Detector Segmentation.  An analysis regarding the improvement of the predictive capacity of time series using segmentation through stationary analysis is included.

The work complies with what is proposed and is a contribution in its area of knowledge.

I only propose (i) to add the data used and (ii) a Discussion Section of the results obtained and the improvements and limitations.

Best regards

Reviewer 3 Report

  1. Segmentation of Time Series based on stationarity has been used for a long time. What is the innovation in this research? You may want to elaborate  a bit on this.
  2. From Algorithm 1 and 2, it seems that only one segment is used in this research. Did I misinterpret them? If not, why?
  3. One disadvantage of using RMSE is that outliers will have huge effect on this metric. Have you tried other metrics? Do they have similar trends?
